# Short-Term Effects of Particulate Matter and Its Constituents on Emergency Room Visits for Chronic Obstructive Pulmonary Disease: A Time-Stratified Case-Crossover Study in an Urban Area

**DOI:** 10.3390/ijerph18094400

**Published:** 2021-04-21

**Authors:** Yii-Ting Huang, Chien-Chih Chen, Yu-Ni Ho, Ming-Ta Tsai, Chih-Min Tsai, Po-Chun Chuang, Fu-Jen Cheng

**Affiliations:** 1Department of Emergency Medicine, Kaohsiung Chang Gung Memorial Hospital, College of Medicine, Chang Gung University, Kaohsiung City 833, Taiwan; kr032961@gmail.com (Y.-T.H.); pancreas133@cgmh.org.tw (C.-C.C.); r223054987@cgmh.org.tw (Y.-N.H.); dada740421@yahoo.com.tw (M.-T.T.); bogy1102@cgmh.org.tw (P.-C.C.); 2College of Medicine, Chang Gung University, Guishan District, Taoyuan City 333, Taiwan; tcmnor@cgmh.org.tw; 3Department of Pediatrics, Kaohsiung Chang Gung Memorial Hospital, College of Medicine, Chang Gung University, Kaohsiung City 833, Taiwan

**Keywords:** air pollution, chronic obstructive pulmonary disease (COPD), constituents, emergency department, particulate matter

## Abstract

Background: PM_2.5_ exposure is associated with pulmonary and airway inflammation, and the health impact might vary by PM_2.5_ constitutes. This study evaluated the effects of increased short-term exposure to PM_2.5_ constituents on chronic obstructive pulmonary disease (COPD)-related emergency department (ED) visits and determined the susceptible groups. Methods: This retrospective observational study performed in a medical center from 2007 to 2010, and enrolled non-trauma patients aged >20 years who visited the emergency department (ED) and were diagnosed as COPD. Concentrations of PM_2.5_, PM_10_, and the four PM_2.5_ components, including organic carbon (OC), elemental carbon (EC), nitrate (NO_3_^−^), and sulfate (SO_4_^2−^), were collected by three PM supersites in Kaohsiung City. We used an alternative design of the Poisson time series regression models called a time-stratified and case-crossover design to analyze the data. Results: Per interquartile range (IQR) increment in PM_2.5_ level on lag 2 were associated with increments of 6.6% (95% confidence interval (CI), 0.5–13.0%) in risk of COPD exacerbation. An IQR increase in elemental carbon (EC) was significantly associated with an increment of 3.0% (95% CI, 0.1–5.9%) in risk of COPD exacerbation on lag 0. Meanwhile, an IQR increase in sulfate, nitrate, and OC levels was not significantly associated with COPD. Patients were more sensitive to the harmful effects of EC on COPD during the warm season (interaction *p* = 0.019). The risk of COPD exacerbation after exposure to PM_2.5_ was higher in individuals who are currently smoking, with malignancy, or during cold season, but the differences did not achieve statistical significance. Conclusion: PM_2.5_ and EC may play an important role in COPD events in Kaohsiung, Taiwan. Patients were more susceptible to the adverse effects of EC on COPD on warm days.

## 1. Introduction

Chronic obstructive pulmonary disease (COPD), a major cause of morbidity and mortality worldwide [1], is defined as persistent airflow limitation without full recovery and chronic inflammation of the lungs [2]. The most significant cause of COPD is cigarette smoking, followed by biomass smoke, especially in developing countries [3]. The model estimated prevalence rate of COPD in 12 Asian countries was around 6.3% (range, 3.5–6.7%) [4]. In 2018, chronic lower respiratory disease was the seventh leading cause of death in men (38.2/100,000) and the ninth in women (14.1/100,000) in Taiwan (https://www.mohw.gov.tw/cp-16-48057-1.html (accessed on 2 March 2019)).

The adverse effects of particulate matter (PM) on cardiovascular and respiratory tract diseases, such as asthma, pneumonia, and ischemic heart diseases, were recently described by several epidemiological studies [5,6,7,8]. PM_2.5_ (PM with an aerodynamic diameter < 2.5 μm, or fine PM) has serious adverse health effects, and epidemiological studies indicate that PM_2.5_ may have a greater health impact than larger air particles (also called PM10, with an aerodynamic diameter < 10 μm) because PM_2.5_ can easily penetrate the smaller portions of the respiratory tract, including the alveoli [6,9]. Toxicological studies also revealed that PM exposure is associated with pulmonary and airway inflammation [10,11]. However, the hazardous effects of PM seem to have regional and seasonal variation. Bell et al. collected data from 202 U.S. cities and found that PM_2.5_, especially in the Northeast region during the winter, was associated with respiratory disease-related hospital admission [12]. Possible explanations for this heterogeneity include the proportion of aged population [13], population density [14], ambient temperature changes [6], and different PM_2.5_ components [15]. The PM_2.5_ components varied between regions, seasons, and weather conditions [16,17], and different PM_2.5_ components may cause different health impacts. Animal studies have highlighted the different health effects of different PM_2.5_ constituents [18,19]. Epidemiologic studies have also observed various health impacts of PM_2.5_ components, such as the influence of sulfate on mortality [20] and ammonium on cardiovascular emergency department (ED) visits [21].

Epidemiologic studies revealed the hazardous effects of PM on COPD, but the seasonal and regional effects remain unclear. Tian et al. showed a positive relationship between PM_2.5_ and COPD-related hospital visits in Beijing, China, especially during the warm season [22], while Hwang et al. demonstrated the relationship between PM_2.5_ and COPD hospital visits in Beijing, China, also during the warm season [23]. One possible reason is that PM_2.5_ constituents vary with seasons and regions. On the other hand, patient characteristics might be another reason for this difference. Previous studies indicated that patients with comorbidities might be at higher risk of out-of-hospital cardiac arrest (OHCA) and cerebral infarction considering the levels of different air pollutants [24,25]. However, few studies have focused on the adverse health effects of PM_2.5_, especially in patients with underlying diseases.

The present study had two specific objectives: evaluate the effects of increased short-term exposure to PM_2.5_ constituents on COPD-related ED visits, and investigate the potential triggering effects of PM_2.5_, especially in patients with underlying diseases.

## 2. Materials and Methods

### 2.1. Study Population

This was a retrospective observational study performed in an urban tertiary medical center with an average of 72,000 ED visits per year in Kaohsiung, Taiwan. Kaohsiung is an industrial city with a population of approximately 2.77 million people and 2952 square kilometers. The study included non-trauma patients aged >20 years who visited the ED with a principal diagnosis of COPD (International Classification of Diseases, Ninth Revision [ICD-9]: 491) between 1 January 2007, and 31 December 2010. Each episode of acute exacerbation of COPD with ED visit is regarded as a “COPD ED visit”, including patients who are hospitalized or discharged from the ED. Medical records from the ED administrative database were reviewed by two trained emergency physicians. Patients’ demographic factors, such as sex, age, and pre-existing comorbidities (including diabetes, hypertension, current smoker, and malignancy) were acquired from the medical records. This study was approved by our hospital’s institutional review board (201900725B0) and was conducted in accordance with the ethical guidelines of the 1964 Declaration of Helsinki and its amendments and comparable ethical standards. Informed consent was not required for this study.

### 2.2. Pollutant and Meteorological Data

Air pollutant monitoring data and meteorological data were acquired from three PM supersites established in Kaohsiung City from 2005 to 2010 by the Taiwanese Environmental Protection Administration. Hourly mass concentrations of PM_2.5_, PM_10_, and the four PM_2.5_ components, including organic carbon (OC), elemental carbon (EC), nitrate (NO_3_^−^), and sulfate (SO_4_^2−^), were collected automatically during the study period. The hourly mass concentrations of PM_2.5_ and PM_10_ were measured using a tapered element oscillation microbalance (Rupprecht and Patashnick 1400a), and those of OC and EC were detected using a Series 5400 Monitor. NO_3_^−^ was detected by a Series 8400N Particulate Nitrate Monitor, and SO_4_^2−^ was detected by a Series 8400S Particulate Sulfate Monitor (Environmental Protection Administration Executive Yuan, R.O.C., Taipei 2010). The percentage of data missing for PM_2.5_ and PM_10_ was less than 1%, and that for each PM_2.5_ constituent was about 1%. We collected the daily average PM and its components of each monitoring supersite as well as the patients’ addresses from their medical records, and the 24-h average levels of these pollutants were computed from the nearest monitoring station. Daily recordings of the mean temperature and mean humidity were also collected from the monitoring stations.

### 2.3. Statistical Method

We used an alternative design of the Poisson time series regression model called the time-stratified and case-crossover design to analyze the data [26,27]. The day of the COPD-related ED visit was defined as lag 0, the day before the episode was lag 1, the day before lag 1 was lag 2, and the like. The effects of environmental conditions on COPD were surveyed and recorded separately for lags 0 to 3. Within-subject comparisons in the case-crossover design were conducted between control periods and cases. Time stratification was performed to select referent days as the days falling on the same day of the week (one case day with three to four control days) within the same month as the index day. The levels of air pollution during the case period were compared with the levels on all referent days. The time stratification method was used to adjust the effects of long-term trends, seasonal variation, and day of the week [28]. Conditional logistic regression was used to estimate the odds ratios (ORs) and 95% confidence intervals (CIs) of COPD associated with PM_2.5_ mass and its constituents. Subgroup analyses were also performed to identify the most susceptible groups based on age, sex, and underlying disease.

Temperature and relative humidity were included in the model as confounding factors in two steps. First, potential non-linear relationships between air temperature, humidity, and COPD exacerbation were determined using the Akaike’s information criterion (AIC) [29]. We used the SAS macro “lgtphcurv9,” which implements the natural cubic spline methodology to fit potential non-linear response curves in logistic regression models for case-control studies [30]. With temperature, the AIC value for the linear model (16,924.132) was better than that for the spline model (16,926.117), while the test of curvature (non-linear relationship) was non-significant (*p* = 0.877). Similarly, with humidity, the AIC value for the linear model (16,913.360) was better than that for the spline model (16,916.296), and the test of curvature was also non-significant (*p* = 0.132). As a result, we used the linear model for the conditional logistic regression analysis.

The ORs were calculated based on the interquartile range (IQR) increments of PM10, PM_2.5_, and its constituents. The significance criterion was set at *p* < 0.05. All statistical analyses were performed using SAS version 9.3 (SAS Institute Inc., Cary, NC, USA).

## 3. Results

During the 7-year study period, 7333 visits to the ED were recorded for patients with COPD. A total of 1219 patients were excluded from the analysis because they did not reside in Kaohsiung City; the remaining 6114 patients were included in our study group. The demographic characteristics of the 6114 patients are presented in Table 1. Of them, 4461 (73.0%) were male and the mean age was 71.7 years. Hypertension (62.3%), diabetes (30.9%), and malignancy (21.0%) were the most frequently recorded underlying diseases.

The daily average temperature, humidity, and concentrations of PM_10_, PM_2.5_, and its constituents in Kaohsiung during the study period are shown in Table 2. The average PM_2.5_ concentration over the study period was 32.7 ± 15.9 μg/m^3^. Among the components of PM_2.5_, SO_4_^2−^, and OC were major constituents, accounting for 9.4 ± 4.8 µg/m^3^ and 8.2 ± 3.7 µg/m^3^, respectively.

Table 3 shows the Spearman correlation coefficients for the air pollutants and weather conditions. PM_2.5_ was highly correlated with PM_10_ (*r* = 0.909, *p* < 0.001), SO_4_^2−^ (*r* = 0.774, *p* < 0.001), and OC (*r* = 0.731, *p* < 0.001) and moderately correlated with NO_3_^−^ (*r* = 0.669, *p* < 0.001), and EC (*r* = 0.568, *p* < 0.001).

Figure 1 shows the year-round estimates of PM_2.5_ and its constituents on COPD ED visits. IQR increases in PM_2.5_ level on lag 2 were associated with increments of 6.6% (95% CI, 0.5–13.0%) in the risk of COPD exacerbation. An IQR increase in EC was significantly associated with an increment of 3.0% (95% CI, 0.1–5.9%) in the risk of COPD exacerbation on lag 0. Meanwhile, an IQR increase in SO_4_^2−^, NO_3_^−^, and OC levels was not significantly associated with COPD.

Figure 2 demonstrates the results of the stratified analysis used to elucidate the effects of PM_2.5_ and EC on COPD according to different seasons, demographic factors, and underlying diseases on lag 2 and lag 0, respectively. As shown in Figure 2a, after the adjustment for temperature and humidity, the risk of COPD exacerbation after exposure to PM_2.5_ was higher in individuals who are currently smoking, with malignancy, or during the cold season, but the differences were not statistically significant. Figure 2b shows that patients were more sensitive to the harmful effects of EC on COPD during the warm days (interaction *p* = 0.019). The risk of COPD exacerbation after EC exposure was higher in individuals with diabetes or malignancy, but the differences were not statistically significant.

An exposure–response relationship between PM_2.5_ level and the risk of COPD was calculated. Figure 3 shows that an elevated PM_2.5_ level was significantly associated with an increased risk of COPD compared with lower levels (Q1, ≤18.9 μg/m^3^); exposure to a Q2 level (PM_2.5_ 18.9–31.6 μg/m^3^) and Q3 level (PM_2.5_ 31.6–43.0 μg/m^3^) were significantly associated with a 10.3% (95% CI, 0.2–21.3%; *p* = 0.046) and 11.8% (95% CI, 0.2–24.8%; *p* = 0.047) increase in the risk of COPD. When PM_2.5_ achieved a Q4 level (>43.0 μg/m^3^), the risk for COPD was significantly elevated to 18.4% (95% CI, 5.2–33.3%; *p* = 0.005) compared with lower levels.

## 4. Discussion

We estimated the effects of PM_2.5_ and its constituents on COPD in this study and found that PM_2.5_ and its constituent, EC, may have a strong relationship with COPD events in Kaohsiung, Taiwan. Furthermore, this study demonstrated that PM_2.5_ exposure is associated with COPD in a dose–response manner, and patients were more sensitive to EC during the warm days.

Many epidemiological studies have demonstrated the association of PM with cardiovascular [7,31,32] and respiratory diseases [6,33,34,35]. However, limited studies have focused on PM_2.5_ constituents and ED visits for respiratory diseases. Darrow et al. found a positive association between OC and ED visits for pediatric pneumonia [36]. Hwang et al. revealed that the PM_2.5_ constituent, nitrate, was associated with asthma-related ED visits [15]. The present study demonstrated that the PM_2.5_ constituent, EC, was positively associated with COPD-related ED visits, especially during warm days.

Some studies found a positive association between PM_2.5_ mass and COPD-related ED visits [22,37], while some studies did not find a statistically significant relationship [38,39]. One possible explanation for this inconsistency may be the differences in lag days collected in the previous studies. Gao et al. found a strong association between PM_2.5_ and COPD-related hospitalization on lags 0 to 7 (average, 7 days). Hwang et al. also found a strong association between PM_2.5_ and COPD-related hospital admission on lags 0 to 5 [23]. However, Peel et al. analyzed the relationship between PM_2.5_ and COPD-related ED visits on lags 0 to 2 [39]; Steib et al. only analyzed the impact of PM_2.5_ on COPD-related ED visits on lags 0 to 2 [38]. Seasonal and regional heterogeneity influenced the health effects of PM. Peng et al. collected data from 100 cities in the US and found a positive association between PM and mortality, especially in the summer in Northeast areas [28]. There might be seasonal and regional variations in PM constituents [16], and different PM constituents contribute to different health impacts [20]. A meta-analysis review collected 12 studies and summarized that PM_2.5_ is positively correlated with COPD-related hospitalization and deaths [40]. The present study’s findings also support the positive association between PM_2.5_ levels and COPD-related ED visits.

Different PM_2.5_ constituents are likely to cause different health hazards. Ueda et al. demonstrated that chloride, EC, and OC had positive associations with mortality due to cardiovascular diseases; in addition, SO_4_^2−^, NO_3_^−^, and ammonium were correlated with mortality from respiratory diseases [20]. Hwang et al. found that PM_2.5_ and its constituent ammonium was positively associated with cardiovascular disease-related ED visits [21]. Jung et al. found that levels of NO_3_^−^, black carbon, and potassium were related to outpatient visits for asthma [41], while OC correlated with pediatric pneumonia and upper respiratory tract infection-related ED visits [36]. There is limited evidence on the PM_2.5_ components and hazardous effects of COPD. Peel et al. collected data from 31 hospitals in Atlanta, where patients visited ED with respiratory diseases. The data of the components of PM_2.5_, including water-soluble metals, EC, OC, acidity, and SO_4_^2−^, were also documented. The results suggested that OC was positively correlated with pneumonia-related ED visits, but the correlation of PM_2.5_ constituents with COPD did not achieve statistical significance [39]. The present study showed that PM_2.5_ and its constituent EC were correlated with COPD-related ED visits. One possible reason for this disparity was the different weather conditions. Previous studies also revealed the interaction effects of air pollution and meteorological factors on health. For example, Hung et al. accumulated data from 39 hospitals and found combination effects of PM with low temperatures on the incidence of the acute coronary syndrome [42], and Cheng et al. revealed a positive association between PM and respiratory diseases with hospital admissions, especially on cold days [43]. Second, the characteristics and habitancy of people in different regions might influence the health effects of air pollution. Zeka et al. revealed that PM was positively associated with daily mortality, and the mortality percentage was higher in densely populated areas [14]. Besides, pollution due to PM_2.5_ when combined with cigarette smoking increases the risk of COPD [44,45]. Differences in the proportion of smokers may also influence the health impacts of PM.

Our study demonstrated that patients were more susceptible to EC, which resulted in more COPD-related ED visits on warm days. The differences in temperature in different areas might impact the health effects of PM. Huang et al. indicated that PM exposure was positively associated with hemorrhagic stroke in warm weather [46]. Cheng et al. indicated that PM_2.5_ was correlated with ED visits for pneumonia with sepsis during the warm season [5]. Besides, a combination of other air pollutants might interact with PM_2.5_ and its constituents, and induce health effects. Xiao et al. demonstrated that the combination of O_3_ and PM_2.5_ constituents including SO_4_^2−^, NO_3_^−^, and ammonium were statistically associated with pediatric pneumonia and bronchitis ED visits [34]. Third, the deposition of EC in the human respiratory system might vary with season [47]. Airway deposition of EC might be associated with airway inflammation and COPD acute exacerbation. Further studies should involve more cities with different climate conditions and more air pollution to clarify the hazardous effect and interactions of different PM_2.5_ constituents on COPD.

Toxicology studies demonstrated that PM exposure may lead to inflammatory reactions in the lung. Huang et al. collected PM from urban traffic areas and conducted an animal study using intratracheal instillation. They found that PM exposure induced T helper 1 (Th1) cells and Th2 cell-related inflammatory reactions, such as inflammatory cytokine release and neutrophil and eosinophil recruitment [48]. Cell studies also revealed that PM_2.5_ exposure might influence human pulmonary epithelial cell viability and induce inflammatory responses, oxidative stress, and cell membrane ruptures [49]. In addition, the health hazards caused by different PM components differ. Repeated airway exposure to extracted solutions of PM_2.5_ may cause inflammation of the respiratory tract and abnormal phospholipase metabolism [19]; organic extracts of PM_2.5_ might cause an increase in DNA adducts and further genotoxicity [50]. The current study found that among the major PM constituents, EC and PM_2.5_ mass, had positive associations with COPD-related ED visits. Previous human studies also showed that EC exposure might reduce children’s forced vital capacity (FVC) and forced expiratory volume in 1 s (FEV1) [51]. For patients with COPD, PM exposure was also found to reduce FEV1 and FVC, which might be related to the acute onset of COPD [52]. The present study also supports that EC exposure might be related to acute COPD exacerbation.

According to the 2019 Taiwan air quality standard revision draft, the PM_2.5_ standard was maintained at 15 µg/m^3^ [53]. However, the health standard announced by the World Health Organization was 10 µg/m^3^, and the United States revised the standard to 12 µg/m^3^ in 2012 [54]. Compared to the lower PM_2.5_ levels (≤10.0 µg/m^3^), Kang et al. indicated that the risk of OHCA increased when PM_2.5_ was higher than 10.0 µg/m^3^, and the risk was even higher when PM_2.5_ was higher than 50 μg/m^3^ [55]. Pan et al. also demonstrated a linear exposure-response relationship between PM_2.5_ and the risk of ST-segment elevation myocardial infarction [7]. Our study results showed that the ORs of COPD increased when the PM_2.5_ concentration was higher than 18.9 µg/m^3^ (Q1) and the ORs increased even more when the PM_2.5_ concentration exceeded 43.0 µg/m^3^ (Q4). In other words, the hazardous effect of PM_2.5_ on COPD seemed to be dose-dependent. Therefore, reducing PM_2.5_ might positively impact public health.

### Study Limitations

There were some limitations to this study. First, it was a retrospective observational study conducted in a medical center, in a tropical region, and in an industrial metropolitan city. Therefore, the results of our research may not be applicable to other locations that are ethnically and meteorologically different. Second, COPD is a clinical diagnosis and the judgment of the emergency physician might influence the diagnosis. Third, we recorded the place where the patient lived, but perhaps the patient spent much more time in the workplace or at school. Forth, factors, such as geographical location, outdoor time, and the use of facial masks and air purifiers may be related to the exposure of the patients to pollutants. Future studies should be conducted in different regions with the investigation of personal air pollution protective equipment and work regions to minimize these limitations.

## 5. Conclusions

PM_2.5_ and EC may play an important role in COPD events in Kaohsiung, Taiwan. Patients were more susceptible to the adverse effects of EC on COPD on warm days. In addition, we hope that the results of this study will motivate the government to pay closer attention to managing air pollution, especially PM_2.5_-based pollution.

## Figures and Tables

**Figure 1 ijerph-18-04400-f001:**
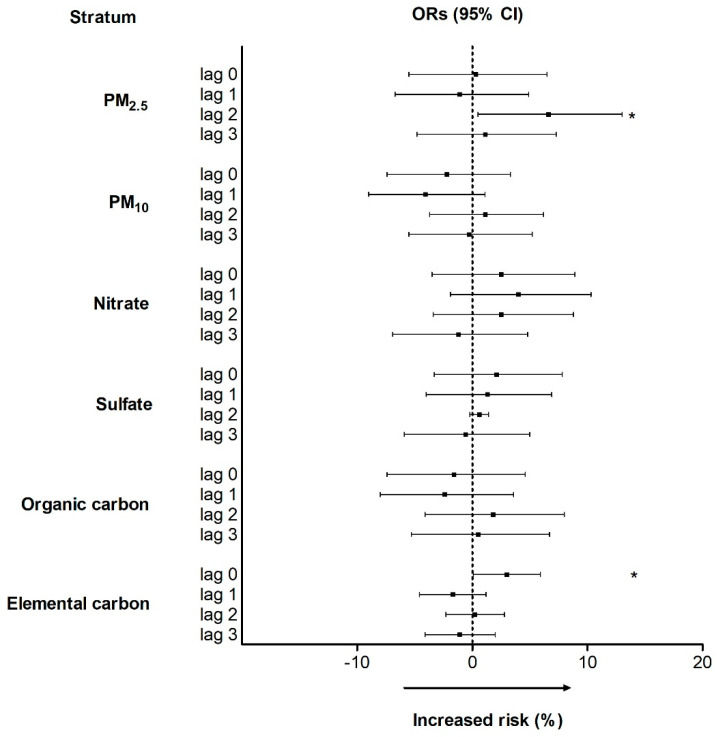
Odds ratios and 95% confidence intervals for COPD ED visits associated with IQR increments in PM_2.5_ and its constituent levels with adjustment for temperature and humidity. COPD, chronic obstructive pulmonary disease; ED, emergency department; IQR, interquartile range. * *p* < 0.05.

**Figure 2 ijerph-18-04400-f002:**
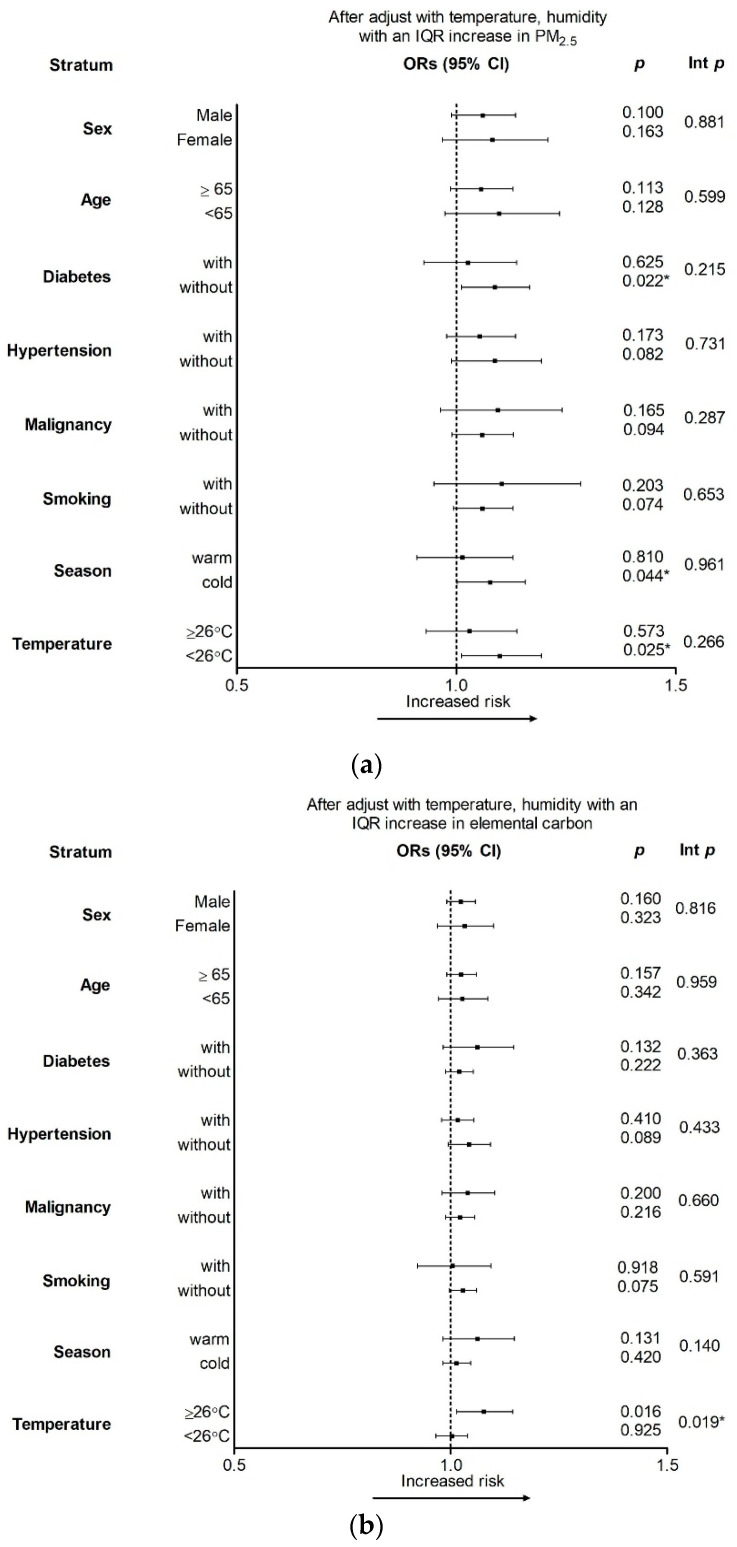
Odds ratios for interquartile range increments in (**a**) PM_2.5_ on lag 2 and (**b**) elemental carbon on lag 0 after the adjustment for temperature and humidity. * *p* < 0.05. COPD, chronic obstructive pulmonary disease; Int p, interaction *p* value.

**Figure 3 ijerph-18-04400-f003:**
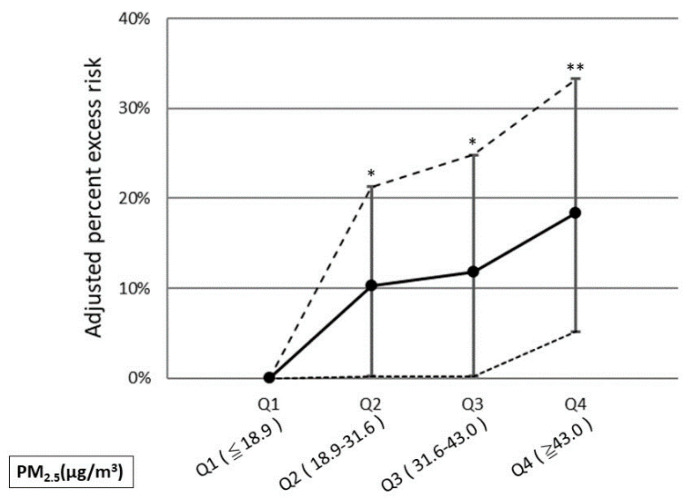
Adjusted risk of COPD according to ambient PM_2.5_ levels. The y-axis represents the percentage of excess risk with 95% confidence intervals. * *p* < 0.05, ** *p* < 0.01.

**Table 1 ijerph-18-04400-t001:** Demographic characteristics of patients (*n* = 6114).

Characteristics	Number	%
Age (mean ± SD)	71.7 ± 12.9	
Male sex	4461	73.0
Diabetes	1889	30.9
Hypertension	3812	62.3
Malignancy	1286	21.0
Current smoker	903	14.8
Warm season	3109	50.9
Warm days (≥26 °C)	3283	53.7

**Table 2 ijerph-18-04400-t002:** Summary statistics for meteorology and air pollution in Kaohsiung, 2007–2010.

Variables	Minimum	Percentiles			Maximum	Mean	IQR
		25%	50%	75%			
PM_2.5_ (µg/m^3^)	6.9	18.9	31.6	43.0	119.5	32.7 ± 15.9	24.1
PM10 (µg/m^3^)	10.7	29.7	46.6	66.9	449.5	50.3 ± 26.3	37.2
Nitrate (µg/m^3^)	0.3	1.4	3.9	6.6	20.7	4.4 ± 3.3	5.2
Sulfate (µg/m^3^)	1.1	5.6	9.1	12.5	33.7	9.4 ± 4.8	6.9
Organic carbon (µg/m^3^)	1.4	5.4	7.5	10.6	27.8	8.2 ± 3.7	5.2
Elemental carbon (µg/m^3^)	0.5	1.5	2.0	2.6	16.5	2.1 ± 0.9	1.1
Temperature (°C)	13.4	22.6	26.5	28.8	31.6	25.5 ± 4.0	6.2
Humidity (%)	44.0	69.0	73.4	77.3	95.3	73.2 ± 7.4	8.3

**Table 3 ijerph-18-04400-t003:** Pearson correlation coefficients between air pollutants and weather conditions in the study period.

Pollutant	PM10	PM_2.5_	Nitrate	Sulfate	Organic Carbon	Elemental Carbon	Temperature (°C)	Humidity (%)
**PM10**		0.909	0.669	0.774	0.731	0.568	−0.493	−0.410
**PM_2.5_**			0.793	0.908	0.822	0.669	−0.504	−0.406
**Nitrate**				0.680	0.833	0.643	−0.580	−0.269
**Sulfate**					0.673	0.592	−0.403	−0.359
**Organic carbon**						0.732	−0.536	−0.377
**Elemental carbon**							−0.376	−0.277
**Temperature (°C)**								0.315
**Humidity (%)**								

## Data Availability

Datasets used in this study will be made available upon request.

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
