# Peer review of "Short-Term Effects of Particulate Matter and Its Constituents on Emergency Room Visits for Chronic Obstructive Pulmonary Disease: A Time-Stratified Case-Crossover Study in an Urban Area"

_ijerph, 2021, doi:10.3390/ijerph18094400_

Round 1

Author Response

Dear editor,

We would like to thank you for your letter dated 06/04/2021 and for providing us with the opportunity to resubmit a revised copy of our manuscript titled “Short-term effects of particulate matter and its constituents on emergency room visits for chronic obstructive pulmonary disease: A time-stratified case-crossover study in an urban area” We would also like to express our gratitude to the reviewers for their positive feedback and helpful comments regarding our manuscript.

We have carefully reviewed the comments and revised the manuscript accordingly. Please find in the pages below our point-by-point responses to the reviewers’ comments. We hope that we have adequately addressed their concerns.

Please let me know if any further information is required. We trust that our manuscript is now suitable for publication, and we look forward to your decision.

Thank you for your consideration.

Sincerely,

Fu-Jen Cheng

Reviewer 2 Report

In the present manuscript, the authors have analysed the effects of different air pollutant on COPD events. The study is relevant but I have the below-mentioned comments:

1. Lack of novelty: Literature is available assessing the impact of different PM particle and air pollutants on COPD exacerbation event. The authors need to describe what are the novel findings of this study?

2. Data is old: This is an observational study from 2007-10 (11 years ago). It would be to include some recent data as then the outcomes of the study will help in formulating health guidelines to different governmental agencies.

3. Previous literature shows that Nitrate and Sulphate levels are associated with COPD ex. events but in the present study, no correlation was found. How do authors explain this?

https://pubmed.ncbi.nlm.nih.gov/31229002/

https://pubmed.ncbi.nlm.nih.gov/23078274/

https://pubmed.ncbi.nlm.nih.gov/27756407/

Minor comments:

  1. In the introduction, section authors have primarily explained PM2.5 but in the study, the effect of Elemental carbon, organic carbon, sulphates, nitrates have been evaluated. A more detailed introduction will surely help the readers.
  2. By COPD events authors mean COPD exacerbation??

Author Response

Dear editor,

We would like to thank you for your letter dated 06/04/2021 and for providing us with the opportunity to resubmit a revised copy of our manuscript titled “Short-term effects of particulate matter and its constituents on emergency room visits for chronic obstructive pulmonary disease: A time-stratified case-crossover study in an urban area” We would also like to express our gratitude to the reviewers for their positive feedback and helpful comments regarding our manuscript.

We have carefully reviewed the comments and revised the manuscript accordingly. Please find in the pages below our point-by-point responses to the reviewers’ comments. We hope that we have adequately addressed their concerns.

Please let me know if any further information is required. We trust that our manuscript is now suitable for publication, and we look forward to your decision.

Thank you for your consideration.

Sincerely,

Fu-Jen Cheng

Department of Emergency Medicine, Kaohsiung Chang Gung Memorial Hospital,

Chang Gung University College of Medicine,

No. 123, Dapi Rd., Niaosong Township, Kaohsiung County 833, Taiwan

Phone No: +886-975-056-646

Fax No: +886-7-7317123-8787

Email Address: [email protected]

Round 2

Reviewer 1 Report

The manuscript was clearly improved.

I still have some minor concerns, that should be clarified:

-It seems that all the admissions were included, even more than one admission for the same patient. This should be clearly stated in the Methods section.

-"COPD incidence" and "risk of COPD" are still cited when I think the authors refer to "COPD exacerbations". Risk of COPD is different from risk of COPD exacebations. The incidence of COPD does not seem to be the objective of this study.

Author Response

Dear editor,

We would like to thank you for your letter dated 15/04/2021 and for providing us with the opportunity to resubmit a revised copy of our manuscript titled “Short-term effects of particulate matter and its constituents on emergency room visits for chronic obstructive pulmonary disease: A time-stratified case-crossover study in an urban area” We would also like to express our gratitude to the reviewers for their positive feedback and helpful comments regarding our manuscript.

We have carefully reviewed the comments and revised the manuscript accordingly. Please find in the pages below our point-by-point responses to the reviewers’ comments. We hope that we have adequately addressed their concerns.

Please let me know if any further information is required. We trust that our manuscript is now suitable for publication, and we look forward to your decision.

Thank you for your consideration.

Sincerely,

Fu-Jen Cheng

Department of Emergency Medicine, Kaohsiung Chang Gung Memorial Hospital,

Chang Gung University College of Medicine,

No. 123, Dapi Rd., Niaosong Township, Kaohsiung County 833, Taiwan

Phone No: +886-975-056-646

Fax No: +886-7-7317123-8787

Email Address: [email protected]

Reviewer reports:

Reviewer 1

Comments and Suggestions for Authors

The manuscript was clearly improved.

I still have some minor concerns, that should be clarified:

-It seems that all the admissions were included, even more than one admission for the same patient. This should be clearly stated in the Methods section.

Response: Thank you for your valuable comment. We have added the following sentence in line 86–88:

“Each episode of acute exacerbation of COPD with ED visit is regarded as a “COPD ED visit”, including patients who are hospitalized or discharged from the ED. Medical records from the ED administrative database were reviewed by two trained emergency physicians.”

-"COPD incidence" and "risk of COPD" are still cited when I think the authors refer to "COPD exacerbations". Risk of COPD is different from risk of COPD exacebations. The incidence of COPD does not seem to be the objective of this study.

Response: Thank you for your valuable comment. Risk of COPD is different from risk of acute exacerbation of COPD. We have added the following sentence in the Method to enhance clarity (line 86–88):

“Each episode of acute exacerbation of COPD with ED visit is regarded as a “COPD ED visit”, including patients who are hospitalized or discharged from the ED. Medical records from the ED administrative database were reviewed by two trained emergency physicians.”

We have also changed the term “COPD incidence” to “risk of COPD exacerbation” in the Abstract, Method, and Results of the manuscript.
